# Estimating the burden of foodborne gastroenteritis due to nontyphoidal *Salmonella enterica*, *Shigella* and *Vibrio parahaemolyticus* in China

**Yan-Jun Li**[1]☉, **Yun-Fan Yang**[1]☉, **Yi-Jing Zhou**[2]☉, **Rong-Hua Zhang**[3]☉, **Cheng-Wei Liu**[4]☉, **Hong Liu**[5]☉, **Xiu-Gui Li**[6]☉, **Wen Chen**[7]☉, **Yan Chen**[1]☉*, **Yong-Ning Wu**[1]☉

1 NHC Key Laboratory of Food Safety Risk Assessment, Research Unit of Food Safety, Chinese Academy of Medical Sciences (No. 2019RU014), China National Center for Food Safety Risk Assessment, Beijing, People's Republic of China, 2 Jiangsu Provincial Center for Disease Control and Prevention, Nanjing, People's Republic of China, 3 Zhejiang Provincial Center for Disease Control and Prevention, Hangzhou, People's Republic of China, 4 Jiangxi Provincial Center for Disease Control and Prevention, Nanchang, People's Republic of China, 5 Shanghai Municipal Center for Disease Control and Prevention, Shanghai, People's Republic of China, 6 Guangxi Zhuang Autonomous Region Center for Disease Control and Prevention, Nanning, People's Republic of China, 7 Sichuan Provincial Center for Disease Control and Prevention, Chengdu, People's Republic of China

☉ These authors contributed equally to this work.
* chenyan@cfsa.net.cn

**Data Availability Statement:** All relevant data are within the paper and Supporting Information file.

## Abstract

To estimate the incidence of foodborne gastroenteritis caused by nontyphoidal *Salmonella enterica*, *Shigella*, and *Vibrio parahaemolyticus* in China, population surveys and sentinel hospital surveillance were implemented in six provinces from July 2010 to July 2011, and a multiplier calculation model for the burden of disease was constructed. The multiplier for salmonellosis and *V. parahaemolyticus* gastroenteritis was estimated at 4,137 [95% confidence interval (CI) 2,320–5,663], and for shigellosis at 4,356 (95% CI 2,443–5,963). Annual incidence per 100,000 population was estimated as 245 (95% CI 138–336), 67 (95% CI 38–92), and 806 (95% CI 452–1,103) for foodborne salmonellosis, shigellosis, and *V. parahaemolyticus* gastroenteritis, respectively, indicating that foodborne infection caused by these three pathogens constitutes an important burden to the Chinese healthcare system. Continuous implementation of active surveillance of foodborne diseases, combined with multiplier models to estimate disease burden, makes it possible for us to better understand food safety status in China.

## Introduction

Foodborne disease is a food safety and public health issue of global concern, causing a heavy burden of disease to society [1–6]. The basic purpose of the disease burden assessment is to determine the influence of various health troubles and potential risks on public health, which is of great significance for the effective allocation of medical resources and the evaluation of

**Funding:** YC received the grants from the National Natural Science Foundation of China (No. 81673175) and CAMS Innovation Fund for Medical Science (CIFMS 2019-I2M-5-024). YJZ received the grant from the Jiangsu Provincial Preventive Medicine Association (No. Y2018084).

**Competing interests:** The authors have declared that no competing interests exist.

the influences of foodborne disease interventions [7]. In the past few decades, many developed countries have carried out studies on the burden of foodborne disease and foodborne pathogens [1–3,5,6,8–10]. The World Health Organization has cooperated with national public health agencies to establish sentinel sites to comprehend the situation in developing countries with a lack of disease burden estimates [11,12]. Research on the burden of foodborne diseases started relatively late in China, with only regional disease burden estimates and a lack of national-level assessment data [13,14]. Before 2010, the data collected through the National Foodborne Diseases Surveillance Network (NFDSN) were basically foodborne disease outbreaks reported by the provincial disease control and prevention agencies [15]. Previous studies have found that traditional passive surveillance systems have obvious underreporting problems, while the situation in China is similar [13,16].

Only a few diseases are clearly related to food, which complicates the assessment of the burden of foodborne diseases. Acute gastrointestinal illness (AGI) is a common manifestation of foodborne diseases, usually caused by food contaminated with pathogens. An important basis for the assessment of the burden of foodborne diseases is the estimation of the incidence of AGI and foodborne pathogen-specific infections in the community. AGI caused by nontyphoidal *Salmonella enterica*, *Shigella*, and *Vibrio parahaemolyticus* is considered to be an important basic information for understanding the burden of foodborne diseases, and foodborne disease outbreaks caused by these pathogens are common reported in China [15]. To calibrate the national surveillance data on foodborne diseases, the China National Center for Food Safety Risk Assessment (formerly the Food Safety Section of the Nutrition and Food Safety Institute of the Chinese Centers for Disease Control and Prevention) launched a pilot program for active surveillance of foodborne diseases in six provinces from 2010 to 2011. Using data from cross-sectional population surveys on AGI and sentinel hospital surveillance on the foodborne specific pathogen obtained in this pilot study, we estimated the burden of foodborne gastroenteritis in China caused by nontyphoidal *S. enterica*, *Shigella*, and *V. parahaemolyticus*.

## Materials and methods

### Population survey

To determine the annual incidence of AGI ($I_{AGI}$), the proportion of AGI cases seeking medical care (Proportion One, $P_1$), and the proportion of AGI cases submitting a stool sample for testing among those seeking medical care (Proportion Two, $P_2$) in the Chinese population, from July 2010 to July 2011, retrospective population surveys were conducted at 20 sentinel sites in six surveillance provinces. The sentinel sites were chosen on the basis of their appropriateness (geographically representative of China, e.g., urban/rural, southern/northern), the wishes of local authorities, and practicability of completing the surveys. Frequency data were weighted to the 2010 Chinese population for age, gender, residence and provinces [17].

Households were randomly selected on a monthly basis from each sentinel site and questionnaires were used to conduct face to face surveys. In the chosen household, one individual next to celebrate his/her birthday was asked about the incidence of AGI in the past 4 weeks. All respondents and parents or guardians of the minors signed the written informed consent before the interview. Proxy respondents were used for all minors aged < 12 years, and for minors aged 12–18 years based on the judgment of parents or guardians. AGI is defined as diarrhea of ≥3 loose stools or severe vomiting with at least one other symptom (abdominal pain/convulsions, fever) within 24 hours, and the respondent has no other known non-infectious causes such as alcoholism, drugs or food allergy, pregnancy. Detailed methods are available in the report of Chen et al. [4].

## Sentinel hospital surveillance

To obtain the number of culture-confirmed cases of nontyphoidal *S. enterica*, *Shigella*, and *V. parahaemolyticus*, sentinel hospital surveillance was carried out during the same period of the population survey. From the six provinces that participated in the population survey, 101 sentinel hospitals were purposefully selected for 12-month surveillance, among which 38 tertiary hospitals, 33 secondary hospitals, and 30 primary hospitals. A primary hospital is defined as a community hospital that provides primary health services; a secondary hospital is defined as a regional hospital that provides comprehensive health services; and a tertiary hospital is defined as a cross-regional hospital that provides comprehensive and specialized health services [18]. The sentinel sites for the hospital surveillance were slightly different from those for population survey, and the details were as follows: (1) Shanghai municipality (Luwan and Qingpu district); (2) Jiangsu province (Suzhou, Taizhou, Wuxi, Xuzhou and Yangzhou prefecture); (3) Zhejiang province (Hangzhou, Huzhou, Jinhua, Quzhou, Shaoxing and Taizhou prefecture); (4) Jiangxi province (Xinyu prefecture); (5) Guangxi province (Baise, Fangchenggang, Guigang, Guilin, Hechi, Liuzhou, Nanning and Wuzhou prefecture); and (6) Sichuan province (Panzhihua prefecture). The population in the sentinel hospital surveillance area ($N_P$) in 2010 was 98,001,891, accounting for about 7.4% of the total permanent population in China (1,332,810,869).

Rectal swabs or fecal specimens from patients who fit the case definition of diarrhea were collected by the sentinel hospitals and were tested for nontyphoidal *S. enterica*, *Shigella*, and *V. parahaemolyticus* by CDC or hospital laboratories. A case of diarrhea was defined as a patient with ≥3 loose stools during any 24 hour period. The stool samples sent to the hospital laboratories are assumed to be representative of the community samples. The China National Center for Food Safety Risk Assessment launched the National Laboratory-based Foodborne Disease Surveillance Network in 2011, but sentinel hospitals are mainly tertiary hospitals, which are completely different from the hospital selection in this pilot study [19]. Therefore, the sentinel hospital surveillance data in this study is limited to the data obtained during the pilot study period from 2010 to 2011.

For *S. enterica* detection, specimens were enriched in selenite broth, followed by surface plating (or plating) on bismuth sulfite agar and xylose-lysine-desoxycholate (XLD) agar (or CHROMagar *Salmonella* agar, Hektoen enteric agar) [20]. For *Shigella* detection, specimens were streaked on XLD agar and MacConkey agar [21]. For *V. parahaemolyticus* detection, specimens were enriched in alkaline peptone water, followed by surface plating (or streaking) on thiosulfate citrate bile salts sucrose agar or CHROMagar *Vibrio* agar [22]. The plates were incubated at 36°C ±1°C for 18–24 hours or 40–48 hours.

## Burden of disease calculation

We multiplied $N_P$, $I_{AGI}$, $P_1$, and $P_2$ to estimate the number of stool specimens submitted in the sentinel hospital surveillance area ($N_{SS}$), then $N_{SS}$ was divided by the actual number of stool samples ($N_{S-1}$ or $N_{S-2}$) to estimate the proportion of laboratory performing test for nontyphoidal *S. enterica* or *V. parahaemolyticus* (Proportion Three, $P_{3-1}$), and *Shigella* (Proportion Three, $P_{3-2}$) (Table 1).

Assuming that the detection sensitivity (Proportion Four, $P_4$) of these laboratories is 87.5% [14], with a range of 85% to 100%. Multiply the reciprocal of the $P_1$, $P_2$, $P_{3-1}/P_{3-2}$, and $P_4$ to obtain the infection underestimation coefficient (Multiplier Total, $M_{T-1}/M_{T-2}$). The pyramid for estimating the burden of pathogen-specific gastroenteritis from culture-confirmed cases is shown in Fig 1.

The number of culture-confirmed cases was multiplied by the above-mentioned infection underestimation coefficient to calculate the number of cases in the surveillance region. PERT distributions with specified minimum, most likely and maximum values were used to model

**Table 1. Steps for the calculation of the frequency of laboratory performing test for pathogens in the surveillance area in China, 2010–2011.**

| Variables | Definition | Data/Formula/distribution | Mean (95% CI [a]) |
|---|---|---|---|
| $N_P$ | Population of sentinel hospital surveillance area ($n$) | 98,001,891 | NA [b] |
| $N_{S-1}$ | No. stool specimens tested for nontyphoidal *Salmonella enteritica* or *Vibrio parahaemolyticus* | 14,048 | NA |
| $N_{S-2}$ | No. stool specimens tested for *Shigella* | 13,341 | NA |
| $I_{AGI}$ | AGI [c] incidence per person-year | PERT (0.16, 0.56, 0.77) [d] | ND [e] |
| $N_{AGI}$ | AGI episodes per year in the sentinel hospital surveillance area ($n$) | $N_P*I_{AGI}$ | 51,777,670 (29,219,641–70,488,691) |
| Proportion One ($P_1$) | Proportion of cases seeking medical care | PERT (36.9%, 56.1%, 76.8%) | ND |
| $N_{MC}$ | Medical consultations for AGI per year in the sentinel hospital surveillance area ($n$) | $N_{AGI}*P_1$ | 29,178,150 (15,401,023–43,947,525) |
| Proportion Two ($P_2$) | Proportion of cases submitting a stool sample for testing among those seeking medical care | PERT (15.6%, 32.7%, 49.9%) | ND |
| $N_{SS}$ | No. stool specimens submitted in the sentinel hospital surveillance area | $N_{MC}*P_2$ | 9,545,200 (4,385,730–16,423,189) |
| Proportion Three ($P_{3-1}$) | Proportion of laboratory performing test for nontyphoidal *Salmonella enterica* or *Vibrio parahaemolyticus* | $N_{S-1}/N_{SS}$ | ND |
| Proportion Three ($P_{3-2}$) | Proportion of laboratory performing test for *Shigella* | $N_{S-2}/N_{SS}$ | ND |
| Proportion Four ($P_4$) | Proportion of laboratory identifying pathogens | PERT (85%, 87.5%, 100%) | ND |
| Multiplier Total ($M_{T-1}$) | The multiplier for nontyphoidal *Salmonella enterica* or *Vibrio parahaemolyticus* | $1/(P_1*P_2*P_{3-1}*P_4)$ | 4,137 (2,320–5,663) |
| Multiplier Total ($M_{T-2}$) | The multiplier for *Shigella* | $1/(P_1*P_2*P_{3-2}*P_4)$ | 4,356 (2,443–5,963) |

[a] CI: Confidence interval.

[b] NA: Not available.

[c] AGI: Acute gastrointestinal illness.

[d] PERT (*a*, *b*, *c*): PERT distribution with minimum value *a*, most likely value *b*, and maximum value *c*.

[e] ND: Not detected.

uncertainty for variables $I_{AGI}$, $P_1$, $P_2$, and $P_4$ [23]. In the supplemental materials, the values behind the minimum, most likely and maximum values reported are presented (S1 Table). The model was performed on @RISK (version 7.6, Palisade, Newfield, N.Y.), a Monte-Carlo simulation software with an estimated 25,000 iterations per time.

The percentage of foodborne pathogen-specific transmission reported by Hald et al. for the Western Pacific Region B region was multiplied by the number of diseases to estimate foodborne nontyphoidal salmonellosis [24], while the proportion reported in the U.S. research was used to estimate foodborne shigellosis and *V. parahaemolyticus* gastroenteritis [3]. The estimated incidences of foodborne gastroenteritis caused by each pathogen were compared with the incidences of foodborne disease outbreak cases calculated from data reported to the China NFDSN.

## Scientific ethics

The study protocol on population survey and sentinel hospital surveillance was approved by the Committee on Human Experimentation of the National Institute for Nutrition and Food Safety, Chinese Center for Disease Control and Prevention. The ethics committee waived the requirement of informed consent for patients with diarrhea.

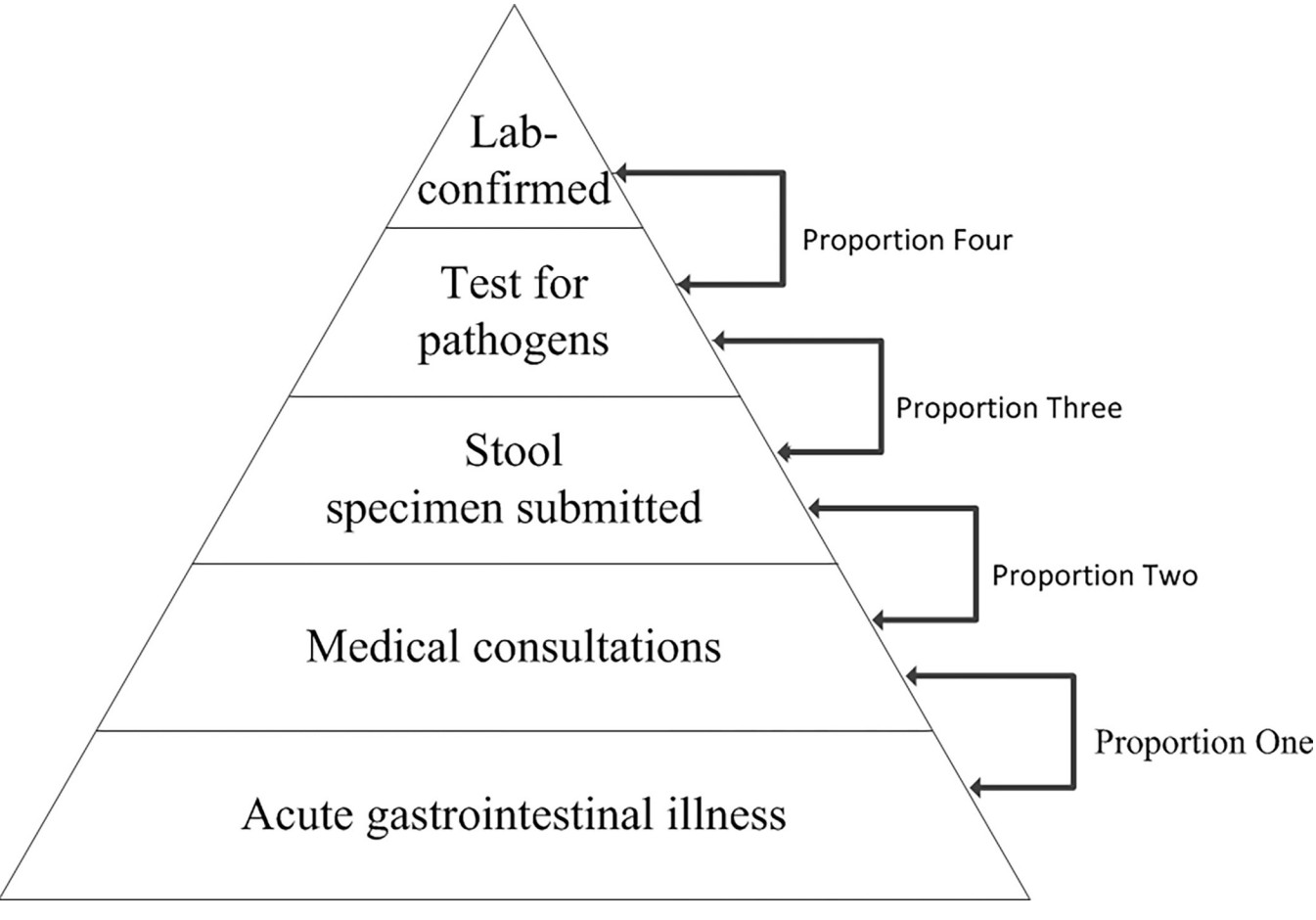

**Fig 1. Estimating the burden of pathogen-specific gastroenteritis from culture-confirmed cases.**

## Results

### Population survey

From July 2010 to July 2011, 39,686 individuals were surveyed (response rate = 93.4%). The median age of the respondents was 47 years (range 0–101 years) and 40% were female. $I_{AGI}$ is 0.56 per person (95% CI 0.56–0.57), ranging from 0.16 and 0.77 for different provinces. Among AGI cases, 56.1% (95% CI 53.6–58.6) saw a doctor, ranging from 36.9% to 76.8% ($P_1$). Among those who saw a doctor, 32.7% (95% CI 29.6–35.8) submitted stool samples, ranging from 15.6% to 49.9% ($P_2$). Detailed results are available in the report of Chen et al. [4].

### Sentinel hospital surveillance

From July 2010 to July 2011, a total of 14,048 stool specimens were collected and tested for nontyphoidal *S. enterica* and *V. parahaemolyticus*. Since *shigella* was not tested for specimen from Jiangxi province, therefore, only 13,341 samples were tested for *Shigella*. Nontyphoidal *S. enterica*, *Shigella*, and *V. parahaemolyticus* were isolated from 102 (0.7%), 42 (0.3%), and 222 (1.5%), respectively, of the AGI cases (Table 2).

There were no isolates in January and February. The monthly isolation rates of nontyphoidal *S. enterica* lay within the scope of 0–2.6%, *Shigella* 0–0.77%, and *V. parahaemolyticus* 0–2.8%. No further research has been done on the isolated bacteria, and no detailed personal information about patients with diarrhea has been collected.

**Table 2. Isolation rate of samples about nontyphoidal Salmonella enterica, Shigella and Vibrio parahaemolyticus in sentinel hospital of six provinces, China, 2010–2011.**

| Province | No. of stool specimens | Nontyphoidal *Salmonella enterica* | | *Shigella* | | *Vibrio parahaemolyticus* | |
|---|---|---|---|---|---|---|---|
| | | No. of isolates | % Isolates | No. of isolates | % Isolates | No. of isolates | % Isolates |
| Shanghai | 4,548 | 19 | 0.4 | 0 | 0.0 | 48 | 1.1 |
| Jiangsu | 2,328 | 22 | 0.9 | 14 | 0.6 | 52 | 2.2 |
| Zhejiang | 4,157 | 8 | 0.2 | 20 | 0.5 | 119 | 2.9 |
| Jiangxi | 707 | 3 | 0.4 | ND[a] | ND | 1 | 0.1 |
| Guangxi | 1,577 | 33 | 2.1 | 5 | 0.3 | 2 | 0.1 |
| Sichuan | 731 | 17 | 2.3 | 3 | 0.4 | 0 | 0.0 |
| Total | 14,048 | 102 | 0.7 | 42 | 0.3 | 222 | 1.6 |

[a] ND: Not detected.

## Burden of disease calculation

Multiplying $N_P$ by $I_{AGI}$, the estimated annual number of AGI cases in the sentinel hospital surveillance area ($N_{AGI}$) is 51,777,673 (95% CI 29,219,641–70,488,691). Multiplying the $N_{AGI}$ by $P_1$, it is estimated that 29,178,150 (95% CI 15,401,024–43,947,526) patients sought medical care ($N_{MC}$). Multiplying the $N_{MC}$ by $P_2$, it is estimated that 9,545,200 (95% CI 4,385,730–16,423,190) people have submitted a stool sample in the sentinel hospital surveillance area (9,740 specimens/100,000 population). The multiplier for salmonellosis and *V. parahaemolyticus* gastroenteritis was estimated at 4,137 [95% confidence interval (CI) 2,320–5,663], and for shigellosis at 4,356 (95% CI 2,443–5,963) (Table 1).

Between July 2010 and July 2011, we estimated 421,980 cases (95% CI 236,655–577,662) of salmonellosis, 182,965 cases (95% CI 102,610–250,466) of shigellosis, and 918,428 cases (95% CI 515,072–1,257,264) of *V. parahaemolyticus* gastroenteritis in the sentinel hospital surveillance area. After referring to the population of the surveillance area, it is estimated that the annual incidence was 431 cases (95% CI 241–589) of salmonellosis, 187 cases (95% CI 105–256) of shigellosis, and 937 cases (95% CI 526–1,283) of *V. parahaemolyticus* gastroenteritis per 100,000 population.

Combined with the percentage of foodborne pathogen-specific transmission [3,24], the annual incidence of foodborne gastroenteritis estimated by our model was 245 cases (95% CI 138–336) for nontyphoidal *S. enterica*, 67 cases (95% CI 38–92) for *Shigella* and 806 cases (95% CI 452–1,103) for *V. parahaemolyticus* per 100,000 population (Table 3).

From 2003 to 2008, according to the foodborne disease outbreaks reported to the China NFDSN by 12 surveillance provinces that include 43% of the Chinese population, there were 3,718 cases due to nontyphoidal *S. enterica*, 581 cases due to *Shigella*, and 9,041 cases due to *V. parahaemolyticus*, and the corresponding incidence rates were 0.11, 0.02, and 0.27 per 100,000 population, respectively [15]. The estimated incidence was much larger than the incidence of reported foodborne disease (Table 3).

## Discussion

This is the first national disease burden report for foodborne pathogens in China derived using a multiplier calculation model. It is estimated that the burden of foodborne gastroenteritis in China was 3,265,387 cases of nontyphoidal salmonellosis, 892,983 cases of shigellosis, and 10,742,456 cases of *V. parahaemolyticus* infection, indicating that foodborne infections

**Table 3. Estimated health burden of nontyphoidal *Salmonella enterica*, *Shigella* and *Vibrio parahaemolyticus* gastroenteritis in China, 2010–2011.**

| Pathogen | No. positive specimens | Estimated positive specimens in the sentinel hospital surveillance area (95% CI [a]) | Estimated No. illness per 100,000 population (95% CI) | Estimated percentage of foodborne transmission [3, 24] [b] | Estimated No. foodborne illness per 100,000 population (95% CI) | Reported No. foodborne outbreak cases per 100,000 population in China [c] |
|---|---|---|---|---|---|---|
| Nontyphoidal *Salmonella enterica* | 102 | 421,980 (236,655–577,662) | 431 (241–589) | 0.57 | 245 (138–336) | 0.11 |
| *Shigella* | 42 | 182,965 (102,610–250,466) | 187 (105–256) | 0.36 | 67 (38–92) | 0.02 |
| *Vibrio parahaemolyticus* | 222 | 918,428 (515,072–1,257,264) | 937 (526–1,283) | 0.86 | 806 (452–1,103) | 0.27 |

[a] CI: Confidence interval.

[b] Estimated percentage of foodborne transmission for nontyphoidal *Salmonella enterica* in gastroenteritis illness in Western Pacific Region B region, and for *Shigella* and *Vibrio parahaemolyticus* in the United States.

[c] The incidences of foodborne disease outbreak cases calculated from data reported to the China National Foodborne Diseases Surveillance Network, from 2003 to 2008.

caused by these three pathogens constitute a huge burden to Chinese healthcare system. Simulated results indicate a significant difference between the estimated incidence and the reported figures of foodborne disease, suggesting that the passive disease surveillance system will inevitably lead to obvious underreporting of foodborne disease cases.

Although there are many differences in data collection and analysis, geographic location, and food culture in the research of different countries, the advantage of using a similar estimation method is that we can sketchily compare the results with those from other countries.

Our estimated foodborne nontyphoidal salmonellosis (245 cases/100,000 population) was higher when compared with Jordan (124 cases/100,000 population) [12] and Japan (199 cases/100,000 population) [10], similar compared with the United Kingdom (220 cases/100,000 population) [25] and Canada (269 cases/100,000 population) [5], and lower compared with the United States (344 cases/100,000 population) [3], Australia (427 cases/100,000 population) [1], African Region (896 cases/100,000 population) [26], Eastern Mediterranean Region (1,610 cases/100,000 population) [26] (Table 4). Compared with previous studies related to China,

**Table 4. Incidence of nontyphoidal *Salmonella enterica*, *Shigella* and *Vibrio parahaemolyticus* foodborne gastroenteritis, by area.**

| Area | Incidence per 100,000 population | | | Reference |
|---|---|---|---|---|
| | Nontyphoidal *Salmonella enterica* | *Shigella* | *Vibrio parahaemolyticus* | |
| China | 245 | 67 | 806 | |
| China | 3,600 | NA [a] | NA | [28] |
| China | 627 | NA | NA | [27] |
| China, Guangdong province | 392 | NA | NA | [14] |
| China, Shanghai | 48 | NA | 183 | [13] |
| African Region | 896 | 523 | NA | [26] |
| Australia | 427 | 2 | 4 | [1] |
| Canada | 269 | 4 | 6 | [5] |
| Eastern Mediterranean Region | 1,610 | 627 | NA | [26] |
| Japan | 199 | NA | 65 | [10] |
| Jordan | 124 | 306 | NA | [12] |
| Thailand | NA | 60 | NA | [29] |
| the United Kingdom | 220 | 27 | <1 | [25] |
| the United States | 344 | 44 | 12 | [3] |

[a] NA: Not available.

the present estimated incidence of foodborne salmonellosis was higher when compared with that reported for Shanghai (48 cases/100,000 population) [13] and lower in comparison with that reported for Guangdong province (392 cases/100,000 population) [14], that determined from a literature review (627 cases/100,000 population) [27], and that reported in a global burden research (3,600 cases/100,000 population) [28] (Table 4).

Our estimate of foodborne shigellosis (67 cases/100,000 population) was higher when compared with the United Kingdom (27 cases/100,000 population) [25], Thailand (60 cases/100,000 population) [29], and lower compared with Jordan (306 cases/100,000 population) [12], African Region (523 cases/100,000 population), and Eastern Mediterranean Region (627 cases/100,000 population) [26] (Table 4).

Our estimate of foodborne *V. parahaemolyticus* infection (806 cases/100,000 population) was much higher than that of the United Kingdom (<1 case/100,000 population) [25], Australia (4 cases/100,000 population) [1], the United States (12 cases/100,000 population) [3] and Japan (65 cases/100,000 population) [10] (Table 4). Previous study has shown that half of outbreaks of *V. parahaemolyticus* infections in China are due to cross-contamination [15]. Seafood in the Chinese market is usually stored at room temperature without cooling facilities, which may lead to a significant increase in the concentration of *V. parahaemolyticus*. Therefore, cross-contamination of ready-to-eat food with raw or uncooked seafood may be the main reason for the high incidence of *V. parahaemolyticus* infection. As the risk of infection with *V. parahaemolyticus* was much higher than the other countries, the health administration in China needs to take effective measures, including the establishment of cold-chain, to prevent and control foodborne infections caused by *V. parahaemolyticus*.

It should be recognized that the uncertainty of this study is higher than that of other countries' estimates. We estimate that there are 4,137 cases of salmonellosis or *V. parahaemolyticus* infection and 4,356 cases of shigellosis in the community for each culture-confirmed case. These differences may be mainly due to the small number of stool specimens tested for specific pathogens. Although prospective cohort studies are more expensive and complicated to implement, they can be combined with laboratory specimen testing to more accurately monitor the incidence of pathogen-specific infections in the region. It is necessary to conduct prospective AGI cohort studies, including the detection of case specimens, to obtain more accurate incidences of foodborne pathogens, so as to reduce the uncertainty of disease burden estimation.

In addition, the limitation of this study is that China has a vast territory and sentinel hospitals only come from six provinces in China, which may lead to underrepresentation. Studies have shown that nontyphoidal salmonellosis is widely distributed in China, shigellosis is mainly distributed in the northwest and inland areas, and *V. parahaemolyticus* infections are mainly distributed in coastal areas [30,31]. The calculation of disease burden is based on linking the number of stool samples from AGI cases in the community to the test results of stool samples from hospital patients. This link is extremely fragile, as the proportion of stool samples tested is only 0.15%. The estimation of community prevalence of infection with individual pathogens from the testing of a tiny proportion of stool samples from cases is likely to be subject to bias and considerable uncertainty. In the present study, the stool samples were obtained from patients seeking medical care for AGI. AGI visits may be the result of a number of factors, such as urban/rural differences, which could bias whether these samples are representative of the community samples. It is recommended to select sentinel hospitals from more provinces, and the number of stool specimens submitted for diagnosis should be proportional to the population. Another limitation of this study is that it did not collect data on the severity of the disease. Although the incidence of disease in different communities is the same, if there are more high-risk individuals in a certain community, the severity of the disease in that community will be higher, which will cause the incidence of disease to be not a good indicator of

the health burden of the community. Therefore, it is extremely necessary to consider the severity of the disease in future studies to improve the current assessment of the burden of foodborne diseases.

As with other disease surveillance, use of burden of foodborne disease estimate results presented in this paper is most effective when compared with past estimates to see the changes through time, under same or similar data collection and calculation format.

## Conclusions

We concluded that foodborne nontyphoidal salmonellosis, shigellosis, and *V. parahaemolyticus* gastroenteritis are important public health problems in China. Simulated results indicate a significant difference between the estimated incidence and the reported incidences of foodborne disease outbreaks, suggesting that the passive disease surveillance system will inevitably lead to obvious underreporting of foodborne disease cases. It is necessary to continuously carry out AGI population surveys and sentinel hospital surveillance to monitor the trend of the burden of foodborne pathogen infection in China. Our estimates of the incidence of nontyphoidal *S. enterica*, *Shigella*, and *V. parahaemolyticus* foodborne gastroenteritis can also provide data references for the global burden of foodborne disease research.

## Supporting information

**S1 Table. The values behind the minimum, most likely and maximum values reported.** (DOCX)

## Acknowledgments

We thank the dedicated partners involved in the population survey and sentinel hospital surveillance for their kindly corporation that made this analysis possible.

## Author Contributions

**Formal analysis:** Yan Chen.

**Funding acquisition:** Yi-Jing Zhou, Yan Chen.

**Investigation:** Yi-Jing Zhou, Rong-Hua Zhang, Cheng-Wei Liu, Hong Liu, Xiu-Gui Li, Wen Chen.

**Methodology:** Yan Chen.

**Project administration:** Yan Chen.

**Supervision:** Yong-Ning Wu.

**Writing – original draft:** Yan-Jun Li, Yun-Fan Yang, Yi-Jing Zhou, Yan Chen.

**Writing – review & editing:** Yan Chen.

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
