## [Decision Letter · Decision Letter 0]

25 Jul 2022

PONE-D-21-34340Estimating the burden of foodborne gastroenteritis due to nontyphoidal Salmonella enterica, Shigella and Vibrio parahaemolyticus in ChinaPLOS ONE

Dear Dr. Chen,

Thank you for submitting your manuscript to PLOS ONE. After careful consideration, we feel that it has merit but does not fully meet PLOS ONE’s publication criteria as it currently stands. Therefore, we invite you to submit a revised version of the manuscript that addresses the points raised during the review process.

The reviewers have raised a number of concerns that need attention. They request additional information on methodological aspects of the study and request some greater detail in the discussion.

We look forward to receiving your revised manuscript.

Kind regards,

Thomas Phillips, PhD

Staff Editor

PLOS ONE

Journal Requirements:

"This study was financially supported by the National Natural Science Foundation of China (No. 81673175), CAMS Innovation Fund for Medical Science (CIFMS 2019-I2M-5-024) and the Jiangsu Provincial Preventive Medicine Association (No. Y2018084)."

"YC received the grants from the National Natural Science Foundation of China (No. 81673175) and CAMS Innovation Fund for Medical Science (CIFMS 2019-I2M-5-024). YJZ received the grant from the Jiangsu Provincial Preventive Medicine Association (No. Y2018084)."

Additional Editor Comments:

'For increased clarity and readability, please alter Figure 1 to use a map which only names/shows provinces where data was collected from. The map present in this article (https://bmcpublichealth.biomedcentral.com/articles/10.1186/1471-2458-13-4560) would be a sufficient replacement and is compliant with our copyright license

Reviewers' comments:

Reviewer's Responses to Questions

**Comments to the Author**

1. Is the manuscript technically sound, and do the data support the conclusions?

Reviewer #1: Yes

Reviewer #2: Yes

2. Has the statistical analysis been performed appropriately and rigorously? 

Reviewer #1: I Don't Know

Reviewer #2: I Don't Know

3. Have the authors made all data underlying the findings in their manuscript fully available?

Reviewer #1: Yes

Reviewer #2: Yes

4. Is the manuscript presented in an intelligible fashion and written in standard English?

Reviewer #1: Yes

Reviewer #2: Yes

5. Review Comments to the Author

Reviewer #1: I found the manuscript in good shape. It may need some editorial correction. In addition, the statistical model may need to be reviewed by more appropriate person.

In general, the manuscript give more insight on the level of the problem in the country. This may be also true in other similar setting in the world, where data is not available.

Reviewer #2: The authors have estimated the burden of foodborne gastroenteritis due to notyphidal Salmonella enteric, Shigella and Vibrio parahaemolyticus infections in China, from sentinel hospital isolate surveillance data and population survey data by using a bayesian statistical modeling method. The surveys and model construction seems to be sound, and although the sample size of stool testing is rather small considering the population of China as the authors also note, the results presented in this paper is useful for health regulators in not only China, but also for other countries.

Below are some comments.

--------

1. Page 5, Line 85

It would be better if you could be more specific in how you chose the population survey sentinel sites, especially what you mean by "appropriateness".

2. Page 8, Line 150

It might be better to explain the reason you used PERT distribution in your model. It would probably be helpful for the readers who are not so familiar with probability distribution.

3. Page 18, Line260-266

Authors have discussed that storage of seafood at room temperature in Chinese market might be the important factor to high Vibrio parahaemolyticus infection estimated and reported in China. It might be more intelligible to add "establishment of cold-chain" in the "effective measures by the government" discussion as one example if it is appropriate.

4. General comment.

Same with other disease surveillance, use of burden of foodborne disease estimate results presented in this paper is most effective when compared with past estimates to see the changes through time, under same or similar data collection and calculation format. I wish the authors to continue their research.

--------

6. PLOS authors have the option to publish the peer review history of their article (what does this mean?). If published, this will include your full peer review and any attached files.

Reviewer #1: No

Reviewer #2: No

---

## [Author Response · Author response to Decision Letter 0]

1 Sep 2022

PONE-D-21-34340

Estimating the burden of foodborne gastroenteritis due to nontyphoidal Salmonella enterica, Shigella and Vibrio parahaemolyticus in China

PLOS ONE

Journal Requirements:

Please remove any funding-related text from the manuscript and let us know how you would like to update your Funding Statement. 

Response: Accepted and removed funding-related text from the manuscript.

Funding Statement is as following:

This study was financially supported by the National Natural Science Foundation of China (No. 81673175), CAMS Innovation Fund for Medical Science (CIFMS 2019-I2M-5-024) and the Jiangsu Provincial Preventive Medicine Association (No. Y2018084).

4. In your Data Availability statement, you have not specified where the minimal data set underlying the results described in your manuscript can be found. PLOS defines a study's minimal data set as the underlying data used to reach the conclusions drawn in the manuscript and any additional data required to replicate the reported study findings in their entirety. All PLOS journals require that the minimal data set be made fully available. 

Response: Accepted and revised the data availability as following:

Yes- all data are fully available without restriction.

5. We note that Figure 1 in your submission contain [map/satellite] images which may be copyrighted. 

Response: Accepted and removed Figure 1.

Reviewers' comments:

1. Page 5, Line 85

It would be better if you could be more specific in how you chose the population survey sentinel sites, especially what you mean by "appropriateness".

Response: Accepted and revised as following:

The sentinel sites were chosen on the basis of their appropriateness (geographically representative of China, e.g., urban/rural, southern/northern) (L87)

2. Page 8, Line 150

It might be better to explain the reason you used PERT distribution in your model. It would probably be helpful for the readers who are not so familiar with probability distribution.

Response: Accepted and revised as following:

Considering the characteristics of the data, a PERT distribution with specified minimum, most likely and maximum values was used to model uncertainty for each proportion. (L151-152)

3. Page 18, Line260-266

Authors have discussed that storage of seafood at room temperature in Chinese market might be the important factor to high Vibrio parahaemolyticus infection estimated and reported in China. It might be more intelligible to add "establishment of cold-chain" in the "effective measures by the government" discussion as one example if it is appropriate.

Response: Accepted and added a sentence as following:

including the establishment of cold-chain (L269)

4. General comment.

Same with other disease surveillance, use of burden of foodborne disease estimate results presented in this paper is most effective when compared with past estimates to see the changes through time, under same or similar data collection and calculation format. I wish the authors to continue their research.

Response: Accepted and added a paragraph as following:

As with other disease surveillance, use of burden of foodborne disease estimate results presented in this paper is most effective when compared with past estimates to see the changes through time, under same or similar data collection and calculation format. (L305-307)

---

## [Decision Letter · Decision Letter 1]

14 Oct 2022

PONE-D-21-34340R1Estimating the burden of foodborne gastroenteritis due to nontyphoidal Salmonella enterica, Shigella and Vibrio parahaemolyticus in ChinaPLOS ONE

Dear Dr. Chen,

Thank you for submitting your manuscript to PLOS ONE. After careful consideration, we feel that it has merit but does not fully meet PLOS ONE’s publication criteria as it currently stands. Therefore, we invite you to submit a revised version of the manuscript that addresses the points raised during the review process.

The revised version has been assessed by a statistical advisor and they have raised a number of concerns that need attention with regard to the methodology of the model used and the statistical analyses presented. These points were raised by the reviewers in the previous round of review but has not been addressed satisfactorily. We would like to give you a further opportunity to address these concerns and those outlined in the statistical advisor's comments.  Please submit your revised manuscript by Nov 28 2022 11:59PM. If you will need more time than this to complete your revisions, please reply to this message or contact the journal office at plosone@plos.org. Please include the following items when submitting your revised manuscript:A rebuttal letter that responds to each point raised by the academic editor and reviewer(s). You should upload this letter as a separate file labeled 'Response to Reviewers'.A marked-up copy of your manuscript that highlights changes made to the original version. You should upload this as a separate file labeled 'Revised Manuscript with Track Changes'.An unmarked version of your revised paper without tracked changes. You should upload this as a separate file labeled 'Manuscript'.

We look forward to receiving your revised manuscript.

Kind regards,

James Mockridge

Staff Editor

PLOS ONE

Reviewers' comments:

Reviewer's Responses to Questions

**Comments to the Author**

1. If the authors have adequately addressed your comments raised in a previous round of review and you feel that this manuscript is now acceptable for publication, you may indicate that here to bypass the “Comments to the Author” section, enter your conflict of interest statement in the “Confidential to Editor” section, and submit your "Accept" recommendation.

Reviewer #3: (No Response)

2. Is the manuscript technically sound, and do the data support the conclusions?

Reviewer #3: Partly

3. Has the statistical analysis been performed appropriately and rigorously? 

Reviewer #3: No

4. Have the authors made all data underlying the findings in their manuscript fully available?

Reviewer #3: Yes

5. Is the manuscript presented in an intelligible fashion and written in standard English?

Reviewer #3: Yes

6. Review Comments to the Author

Reviewer #3: The response to one reviewer wanting a more detailed description of the PERT model included the comment by the investigators that, considering the characteristics of the data, a PERT distribution with specified minimum, most likely and maximum values was used to model uncertainty for each proportion. (L151-152). This was a poorly presented partial response to the larger concern about the general statistical presentation.

Table 1 gives a good descriptive summary of some of the results including the PERT model. However, the detail of the third column (Data, formula, distribution) could be enhanced. Presumably, the PERT (A,B,C) inclusions assume that the A, B and C are these min, most likely, and max values mentioned above. Please correct this reviewer if I am in error. Otherwise, the last column appears to summarize the results being sought. The investigators have to pay attention to communicating these results in a way that is interpretable by a non statistical reader. The entire manuscript should be reviewed and edited to assure clarity of the statistical presentation.

7. PLOS authors have the option to publish the peer review history of their article (what does this mean?). If published, this will include your full peer review and any attached files.

Reviewer #3: No

---

## [Author Response · Author response to Decision Letter 1]

18 Oct 2022

Reviewer #3’s Comments: 

1-The response to one reviewer wanting a more detailed description of the PERT model included the comment by the investigators that, considering the characteristics of the data, a PERT distribution with specified minimum, most likely and maximum values was used to model uncertainty for each proportion. (L151-152). This was a poorly presented partial response to the larger concern about the general statistical presentation.

RESPONSE. Thanks for raising this important issue. Your suggestions have greatly improved the paper. In the revised manuscript, we have clarified the PERT distribution as following.

PERT distributions with specified minimum, most likely and maximum values were used to model uncertainty for variables IAGI, P1, P2, and P4. (L149-151)

2-Table 1 gives a good descriptive summary of some of the results including the PERT model. However, the detail of the third column (Data, formula, distribution) could be enhanced. Presumably, the PERT (A,B,C) inclusions assume that the A, B and C are these min, most likely, and max values mentioned above. Please correct this reviewer if I am in error. Otherwise, the last column appears to summarize the results being sought. The investigators have to pay attention to communicating these results in a way that is interpretable by a non statistical reader. The entire manuscript should be reviewed and edited to assure clarity of the statistical presentation.

RESPONSE. We are grateful for these insightful questions. In the revised manuscript, we have added the following foodnote to Table 1.

d PERT (a, b, c): PERT distribution with minimum value a, most likely value b, and maximum value c. (L166)

---

## [Decision Letter · Decision Letter 2]

24 Oct 2022

Estimating the burden of foodborne gastroenteritis due to nontyphoidal Salmonella enterica, Shigella and Vibrio parahaemolyticus in China

PONE-D-21-34340R2

Dear Dr. Chen,

We’re pleased to inform you that your manuscript has been judged scientifically suitable for publication and will be formally accepted for publication once it meets all outstanding technical requirements.

Kind regards,

Nadim Sharif, M.Sc.

Academic Editor

PLOS ONE

Additional Editor Comments (optional):

Reviewers' comments:

Reviewer's Responses to Questions

**Comments to the Author**

1. If the authors have adequately addressed your comments raised in a previous round of review and you feel that this manuscript is now acceptable for publication, you may indicate that here to bypass the “Comments to the Author” section, enter your conflict of interest statement in the “Confidential to Editor” section, and submit your "Accept" recommendation.

Reviewer #3: All comments have been addressed

2. Is the manuscript technically sound, and do the data support the conclusions?

Reviewer #3: (No Response)

3. Has the statistical analysis been performed appropriately and rigorously? 

Reviewer #3: (No Response)

4. Have the authors made all data underlying the findings in their manuscript fully available?

Reviewer #3: (No Response)

5. Is the manuscript presented in an intelligible fashion and written in standard English?

Reviewer #3: (No Response)

6. Review Comments to the Author

Reviewer #3: (No Response)

7. PLOS authors have the option to publish the peer review history of their article (what does this mean?). If published, this will include your full peer review and any attached files.

Reviewer #3: No
